# Deep processing of structured data

## Abstract

We construct a general unified framework for learning representation of structured data, i.e. data which cannot be represented as the fixed-length vectors (e.g. sets, graphs, texts or images of varying sizes). The key factor is played by an intermediate network called SAN (Set Aggregating Network), which maps a structured object to a fixed length vector in a high dimensional latent space. Our main theoretical result shows that for sufficiently large dimension of the latent space, SAN is capable of learning a unique representation for every input example. Experiments demonstrate that replacing pooling operation by SAN in convolutional networks leads to better results in classifying images with different sizes. Moreover, its direct application to text and graph data allows to obtain results close to SOTA, by simpler networks with smaller number of parameters than competitive models.

## 1 Introduction

Neural networks are one of the most powerful machine learning tools. However, in its basic form, neural networks can only process data represented as vectors with fixed size. Since very often the real data cannot be directly represented in this form (e.g. graphs, sequences, sets), one needs to construct a data specific architecture to process them.

In natural language processing (NLP), documents are represented as sequences with different lengths. Typical approach relies on embedding words into a vector space (Mikolov et al., 2013; Pennington et al., 2014) and processing them with use of recurrent (Bahdanau et al., 2014) or convolutional neural network (Kim, 2014). Since the number of words in a document varies, it is impossible to directly apply fully connected layers afterwards, e.g. for classification of machine translation (Sundermeyer et al., 2012; Cho et al., 2014). To do so, an intermediate pooling operator (sum or max) or zero padding are used to produce a fixed-length vectors (Kalchbrenner & Blunsom, 2013; Sutskever et al., 2014). Alternatively, one may sum all word embeddings before applying recurrent network to make inputs equally sized, which is a basis of typical bag-of-words representation. Both approaches have difficulties in handling context of long documents, which was the reason of introducing attention models (Bahdanau et al., 2014).

Analogically, in computer vision, fully convolutional networks can process inputs of arbitrary shapes, but produced outputs have different sizes as well (Ciresan et al., 2011; Karpathy & Fei-Fei, 2015). This is acceptable for image segmentation (Ronneberger et al., 2015) or inpainting (Iizuka et al., 2017), but to classify such images, we need to produce fixed-length output vectors. This can be obtained by scaling images at preprocessing stage (He et al., 2015) or by applying pooling layer at the top of the network (Long et al., 2015; Maggiori et al., 2016).

Clearly, every new data type causes nontrivial problems when processed with neural network. This is observed in the case of graphs (Brin & Page, 1998; Frasconi et al., 1998). One option is to extract predefined features, e.g. fingerprints in cheminformatics (Klekota & Roth, 2008; Ewing et al., 2006), which however requires extensive domain knowledge. Alternatively, graph recurrent or convolutional networks can be used, but then we need to deal again with outputs of different sizes (Scarselli et al., 2009; Li et al., 2015; Hagenbuchner et al., 2003; Tsoi et al., 2003).

The aim of this paper is to construct a general approach for learning representation of structured objects. We replace pooling operation by our set aggregation network (SAN), which maps an input set (our representation of structured object) to a fixed-length vector in high dimensional latent space, see Figure 1. Our main theoretical result (Theorem 3.1) shows that for latent space of sufficiently large dimension, SAN can learn a unique representation of every input set, which justifies this ap-

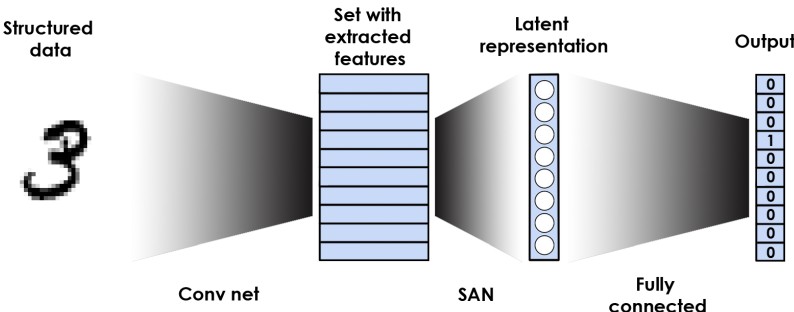

Figure 1: Diagram of proposed architecture. SAN is an intermediate network which is responsible of learning vector representation of structured data.

proach from a theoretical perspective. We show that SAN can be easily implemented in typical deep learning frameworks. In consequences it is a promising alternative to typical pooling operation used in a wide range of architectures, such as recurrent or convolutional networks.

We conduct extensive experiments on different types of structured data (Section 4). Its direct application to text and graph data allows to obtain results close to SOTA, by simpler networks with smaller number of parameters than competitive methods. Moreover, the combination of SAN with convolutional network outperforms typical pooling operation in the case of classifying images with different sizes (section 5).

For the convenience of the reader, we describe the content of the paper. In the following section, we recall a typical way for processing structured data and next describe our approach. In Section 3 we give a theoretical justification of our model. Basic capabilities of SAN on various types of structured data is presented in Section 4. Its practical usefulness is verified in the case of classifying images of various sizes in section 5.

## 2  SET AGGREGATION NETWORK: CONSTRUCTION

Suppose that we want to process a structured data $X = (x_i)_i \subset \mathbb{R}^D$ by a neural network. It can be for example a sequence of word embeddings, image with missing attributes or some graph structure. Typical approach relies on creating two networks combined with an intermediate pooling layer:

$$X = (x_i)_i \overset{\Psi}{\to} (\Psi x_i)_i \overset{\text{Pool}}{\to} \text{Pool}\{\Psi(x_i) : i\} \overset{\Phi}{\to} \mathbb{R}^N.$$

The first network $\Psi : \mathbb{R}^D \to \mathbb{R}^K$ is usually a convolutional or recurrent network, which transforms elements of $X$ sequentially one-by-one and produces a set (or sequence) of vectors. Next, a pooling layer summarizes the response of $\Psi$ and returns a single $K$-dimensional vector. A pooling layer, commonly implemented as sum or maximum operation, gives a vector representation of structured object. Finally, a network $\Phi : \mathbb{R}^K \to \mathbb{R}^N$ maps resulting representation to the final output[1].

A basic problem with the above pipeline lies in the pooling layer. This layer is often a bottleneck and can lead to the lose of meaningful information contained in the whole set structure. Our goal is to learn a representation of a set by replacing classical pooling operation by a specific neural network – set aggregation network (SAN), see Figure 1.

SAN is a neural network $T : \mathbb{R}^K \to \mathbb{R}^M$, which transforms a set[2] of $K$-dimensional vectors into a single vector in $M$-dimensional latent space. The output neurons of $T$ give a representation of a given set by aggregating information contained in a whole set and thus will be referred to as aggregative neurons.

---

[1]One can also create simpler architecture, where the sum or average pooling is applied directly to the set of word embeddings, which is analogical to the use of bag-of-words representation.

[2]One can always add an index coordinate to every vector to map a sequence to a set.

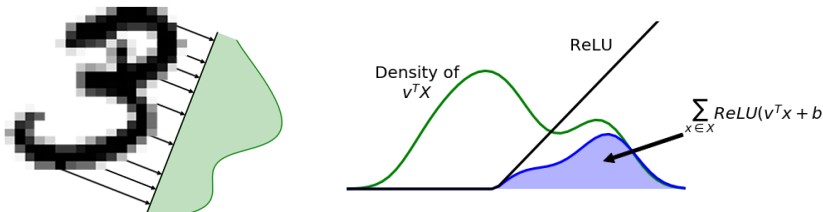

Figure 2: The idea of our approach is to aggregate information from projections of a set onto several one-dimensional subspaces (left). Next non-linear activation function is applied to every set element and the results are aggregated (right).

Our basic idea comes from computer tomography, where Radon transform (Radon, 1986; van Ginkel et al., 2004) is used to reconstruct a function (in practice the 2D or 3D image) from the knowledge of its integration over all one-dimensional lines. A similar statement is given by the Cramer-Wold Theorem (Cramér & Wold, 1936), which says that for every two distinct measures one can find a one-dimensional projection which discriminates them. This implies that without loss of information we can process the set $X \subset \mathbb{R}^K$ through its all one-dimensional projections $v^T X \subset \mathbb{R}$, where $v \in \mathbb{R}^K$ is arbitrary.

In consequence, we reduced the question of representing a multidimensional set to the characterization of one-dimensional sets. One can easily see (for a more general statement see Section 3) that the one-dimensional set $S \subset \mathbb{R}$ can be retrieved from the knowledge of aggregated ReLU on its translations: $b \to \sum_i \mathrm{ReLU}(s_i + b)$, see Figure 2. Summarizing the above reasoning, we obtain that the full knowledge of a set $X \subset \mathbb{R}^K$ is given by the scalar function

$$\mathbb{R}^K \times \mathbb{R} \ni (v, b) \to \sum_i \mathrm{ReLU}(v^T x_i + b).$$

Now, given $M$ vectors $v_i \in \mathbb{R}^K$ and biases $b_i \in \mathbb{R}$, we obtain the fixed-size representation of the set $X \subset \mathbb{R}^K$ as a point in $\mathbb{R}^M$ given by

$$[\sum_i \mathrm{ReLU}(v_1^T x_i + b_1), \dots, \sum_i \mathrm{ReLU}(v_M^T x_i + b_M)] \in \mathbb{R}^M.$$

We arrived at the formal definition of aggregative neuron:

**Definition 2.1.** *Let $\tau : \mathbb{R}^K :\to \mathbb{R}$ be a given neuron's activation function (where as the default we take the ReLU). We extend the use of $\tau$ for finite sets $X = (x_i)_i \subset \mathbb{R}^K$ by the formula:*

$$\tau(X) := \sum_i \tau(x_i). \tag{1}$$

Every aggregative neuron computes a summarized activation of all set elements. A composition of $M$ aggregative neurons gives $M$-dimensional vector representation of the input set. Parameters of aggregation neurons are trainable and thus we allow for learning a representation of structured objects. The final equation (1) in SAN is quite simple and can be easily added to neural network architecture since it reflects simultaneously the structure of neural networks and model natural representation of the set of points.

**Remark 2.1.** *Observe that max-pooling is a special case of SAN. Clearly, for non-negative scalar data $X = (X_i) \subset \mathbb{R}$ and function $\tau_p(x) = x^p$, we have:*

$$\tau^{-1}(\sum_i \tau(x_i)) \to \max_i(x_i) \, , \, as \, p \to \infty.$$

*To obtain a maximum, we use $\tau$ as a activity function in aggregative neuron, which is followed by a layer with its inverse. By extending this scheme, we can get a maximum value for every coordinate. Additionally, to deal with negative numbers, we first take the exponent followed by logarithm after the aggregation.*

We explain how to implement SAN in typical deep learning frameworks. Let $\mathcal{X} = \{X_1, \ldots, X_n\}$ be a batch, which consists of finite sets $X_i \subset \mathbb{R}^K$. While typical networks process vector by vector, SAN needs to compute a summary activation over all elements of $X_i$ before passing the output to the subsequent network.

To deal with sets of different sizes in one batch, we first transform every set $X_i$ to the set of tuples $T_i = \{(x, 1) : x \in X_i\}$. Next, we complement every $T_i$ to have the same sizes by adding zero tuples $(x, 0)$, where $x \in \mathbb{R}^K$ can be an arbitrary vector. The second element of each tuple indicates whether a vector appears in a set or is a dummy vector. The set of tuples $T_i$ can be understood as a uniform discrete measure with a support defined on a given set $X_i$.

After the above operation every element from a batch is represented as a matrix of the same size. To compute the value of aggregative neuron of $X_i$, we apply 1D convolution to the first element of each tuple from $T_i$ and multiply the outputs by their weights. Taking a sum, we get the activation of aggregative neuron $\tau$ for $X_i$:

$$\tau(X_i) = \sum_{x \in X_i} 1 \cdot \tau(x_i) = \sum_{(x,f) \in T_i} f \cdot \tau(x).$$

These operations are applied to every element $X_i$ of a batch. Summarizing, adding SAN to existing network can be implemented in a few lines of codes.

## 3 Theoretical analysis

Our main theoretical result shows that SAN is able to uniquely identify every input set, which justifies our approach from theoretical perspective.

To prove this fact, we will use the UAP (universal approximation property). We say that a family of neurons $\mathcal{N}$ has UAP if for every compact set $K \subset \mathbb{R}^D$ and a continuous function $f : K \to \mathbb{R}$ the function $f$ can be arbitrarily close approximated with respect to supremum norm by $\mathrm{span}(\mathcal{N})$ (linear combinations of elements of $\mathcal{N}$).

We show that if a given family of neurons satisfies UAP, then corresponding aggregation neurons allows to distinguish any two finite sets:

**Theorem 3.1.** *Let $X, Y$ be two discrete sets in $\mathbb{R}^D$. Let $\mathcal{N}$ be a family of functions having UAP.*

*If*

$$\tau(X) = \tau(Y) \text{ for every } \tau \in \mathcal{N}, \tag{2}$$

*then $X = Y$.*

*Proof.* Let $\mu$ and $\nu$ be two measures representing sets $X$ and $Y$, respectively, i.e. $\mu = \mathbf{1}_X$ and $\nu = \mathbf{1}_Y$. We show that if $\tau(X) = \tau(Y)$ then $\mu$ and $\nu$ are equal.

Let $R > 1$ be such that $X \cup Y \subset B(0, R-1)$, where $B(a, r)$ denotes the closed ball centered at $a$ and with radius $r$. To prove that measures $\mu, \nu$ are equal it is sufficient to prove that they coincide on each ball $B(a, r)$ with arbitrary $a \in B(0, R-1)$ and radius $r < 1$.

Let $\phi_n$ be defined by

$$\phi_n(x) = 1 - n \cdot d(x, B(a, r)) \text{ for } x \in \mathbb{R}^D,$$

where $d(x, U)$ denotes the distance of point $x$ from the set $U$. Observe that $\phi_n$ is a continuous function which is one on $B(a, r)$ an and zero on $\mathbb{R}^D \setminus B(a, r + 1/n)$, and therefore $\phi_n$ is a uniformly bounded sequence of functions which converges pointwise to the characteristic funtion $\mathbf{1}_{B(a,r)}$ of the set $B(a, r)$.

By the UAP property we choose $\psi_n \in \mathrm{span}(\mathcal{N})$ such that

$$\sup_{x \in B(0,R)} |\phi_n(x) - \psi_n(x)| \leq 1/n.$$

Thus $\psi_n$ restricted to $B(0, R)$ is also a uniformly bounded sequence of functions which converges pointwise to $\mathbf{1}_{B(a,r)}$. Since $\psi_n \in \mathcal{N}$, by (2) we get

$$\sum_{x \in X} \mu(x) \psi_n(x) = \sum_{y \in Y} \nu(y) \psi_n(y).$$

Table 1: AUC scores for classifying active chemical compounds .

|  | ECFP4 | KlekFP | Mordred | JTVAE | set of atoms |
|---|---|---|---|---|---|
| Random Forest | 0.726 | 0.604 | 0.493 | 0.527 | - |
| Extra Trees | 0.713 | 0.667 | 0.501 | 0.470 | - |
| k-NN | 0.532 | 0.569 | 0.546 | 0.521 | - |
| Dense network | 0.702 | 0.574 | 0.593 | 0.560 | - |
| SAN | - | - | - | - | 0.643 |

Now by the Lebesgue dominated convergence theorem we trivially get

$$\sum_{x \in X} \mu(x)\psi_n(x) = \int_{B(0,R)} \psi_n(x)d\mu(x) \to \mu(B(a,r)),$$
$$\sum_{y \in Y} \nu(y)\psi_n(y) = \int_{B(0,R)} \psi_n(x)d\nu(x) \to \nu(B(a,r)),$$

which completes proof. □

One could generalize the formula of aggregation neurons (1) to the case of arbitrary finite measures $\mu$ by

$$\tau(\mu) = \int \tau(x)d\mu(x).$$

For example, it can be used to process data with missing or uncertain attributes (Smieja et al., 2018; Dick et al., 2008; Williams et al., 2005). Although analogical theoretical result holds in this case, analytical formulas for aggregative neurons can be obtained only for special cases. Smieja et al. (2018) calculated analytical formulas for measures represented by the mixture of Gaussians and aggregative neurons parametrized by ReLU or RBF placed at the first hidden layer.

## 4 EXPERIMENTS: PROOF OF CONCEPT

In this section, as a proof of concept, we show that SAN is capable of processing various types of structured data: graphs, text and images. We are more interested here in demonstrating its wide applicability than obtaining state-of-the-art performance on particular problem.

### 4.1 DETECTING ACTIVE CHEMICAL COMPOUNDS FROM GRAPH STRUCTURES

Finding biologically active compounds is one of basic problems in medical chemistry. Since the laboratory verification of every molecule is very time and resource consuming, the current trend is to analyze its activity with machine learning approaches. For this purpose, a graph of chemical compound is typically represented as a fingerprint, which encodes predefined chemical patterns in a binary vector (Figure 3). Since a multitude of chemical features can be taken into account, various types of fingerprints have been created.

As a simple alternative to chemical fingerprints, we encoded a molecule by a set of its atoms in 2D space. More precisely, a vector representing the atom contains its

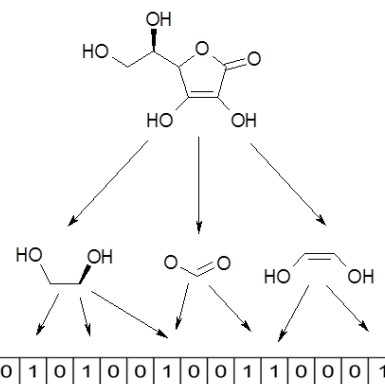

Figure 3: Example of a chemical compound represented by graph and its lossy conversion to fingerprint.

Table 2: Accuracy scores obtained for text data.

| dataset | CNN | SAN |
|---------|--------|--------|
| MR | 0.7623 | **0.7646** |
| imdb | **0.8959** | 0.8480 |

two spatial coordinates (scaled to $(-1, 1)$)
and atom type encoded in one-hot style. This set was the input to SAN, which represented every compound in 100-dimensional latent space. This was followed by three dense layers.

For a comparison, we used four fingerprint representations: (a) Extended-Connectivity Fingerprints (ECFP4) Rogers & Hahn (2010) with 1024 attributes, (b) Klekota-Roth Fingerprint (KlekFP) Klekota & Roth (2008) with 4086 attributes, (c) Mordred fingerprint Moriwaki et al. (2018) with 1613 attributes, (d) representation based on latent space of Junction-Tree Variational Auto-encoder (JTVAE) Jin et al. (2018) with 56 attributes. To learn from each fingerprint, we considered four machine learning algorithms: (a) Random Forest (b) Extra Trees (c) k-NN (d) neural network with four dense layers.

We used a data set of 6 131 chemical compounds which act on A2A receptor ligands, that was taken from ChEMBL database Gaulton et al. (2016). The results presented in Table 1 show that ECFP4 is the best representation for determining chemical compound activity. All methods except k-NN obtained the highest scores on this fingerprint. More interesting thing is that SAN outperformed analogical dense network applied to any other fingerprint representation. This shows that SAN was able to learn better representation using a raw graph structure than handcrafting chemical features by the experts. To obtain even more powerful representation, one could preprocess a graph structure to a set using dedicated convolutional or recurrent networks.

## 4.2 TEXT CLASSIFICATION FROM WORD EMBEDDINGS

Text documents are usually represented as sequences of word embeddings and processed by convolutional or recurrent networks. Since the number of words in documents varies, the outputs are aggregated using pooling to a vector of fixed-size for a classification task. In this experiment, we compare the above complex approach with a direct application of SAN to the sequence of word embeddings (with no preprocessing network).

We used large imdb Maas et al. (2011) and small MR [3] movie review data sets for classification into positive and negative categories. As a reference, we used CNN model following the setting presented in Kim (2014). This model consists of a trainable embedding layer, one-dimensional convolutional layers with 100 filters for each of the kernel size in the set $\{2, 3, 4\}$. The output of each convolution is max-reduced to form a 300 dimensional vector, which is passed to the dense layer with a softmax activation. The model uses dropout on the penultimate layer (see Appendix B for details).

In our model, we replaced convolution and max-pooling layers with a single SAN layer. In consequence, the number of parameters was significantly reduced. To allow for the SAN to account for the feature locality, we concatenate to the word embedding vector the position of the word in the sentence normalized to $(-1, 1)$.

The results presented in Table 2 show that SAN obtained similar results to CNN on MR data set. For a larger imdb data set, SAN gave slightly lower results. This suggests that aggregation neurons are able to preserve meaningful information contained in structured data. However, this information cannot be easily utilized without extracting local features (e.g. using convolutional layers or adding more parameters). A possible remedy is to use SAN after convolutional layers instead of max-pooling, which will be verified in Section 5.

---

[3]https://www.cs.cornell.edu/people/pabo/movie-review-data/

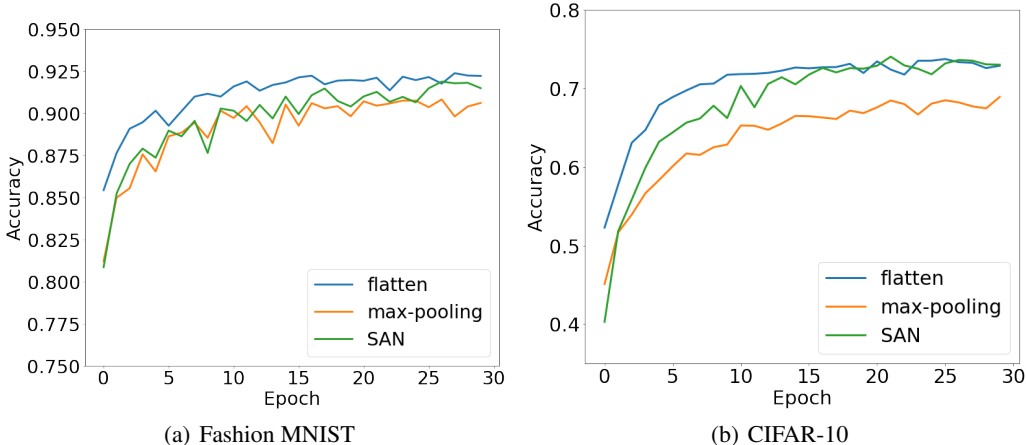

(a) Fashion MNIST          (b) CIFAR-10

Figure 4: Classification accuracy on images with a fixed resolution.

## 4.3 IMAGE CLASSIFICATION

In previous sections, we applied SAN directly to basic representation of structured data (graph representing chemical compounds or sequence of word embeddings). In this experiment, we verify how SAN works in a combination of preprocessing convolutional network.

We considered a classification task on Fasion-MNIST (Xiao et al., 2017) and CIFAR-10 (Krizhevsky & Hinton, 2009). Images in each data set have the same resolution and, in consequence, no pooling operation is needed to apply fully connected layers. Nevertheless, we would like to verify how much information we lose when we apply SAN for a possible aggregation compared to the classical approaches.

We used a neural networks with 3 convolutional layers and max pooling between them (see Appendix A for details of architecture). To aggregate resulted tensor we considered three variants:

- **flatten**: We flattened a tensor to preserve the whole information from previous network.
- **max-pooling**: We applied max pooling along spatial dimensions (width and hight of a tensor) to reduce a dimensionality. In consequence, we obtained a vector of the size equal the depth of a tensor.
- **SAN**: We used SAN as an alternative aggregation. The resulting tensor was treated as a set of vectors with sizes equal the depth of a tensor. Moreover, the (normalized) indices were added to every vector to preserve a local structure.

In each case, the aggregation step was followed by two dense layers. The number of parameters used in **flatten** was higher than in other architectures, because no aggregation was used. To prevent from the overfitting, we used dropout with $p = 0.5$. On the other hand, **max-pooling** had less parameters than **SAN**. To make both approaches comparable we added an extra dense layer to **max-pooling**.

The results presented in Figure 4 show that **SAN** preserved the whole information from convolutional network and gave comparable results to **flatten**. On the other hand, **max-pooling** was unable to utilize this information properly, which is especially evident on CIFAR-10.

## 5 EXPERIMENTS: CLASSIFYING IMAGES WITH VARIED SIZES

Previous experiments presented basic capabilities of SAN. In this section, we present a real-world experiment, where SAN was combined with convolutional network to classify images with varied resolutions.

Most classification models assume that input images are of the same sizes. If this is not the case, we are forced to scale images at preprocessing stage or use pooling operation as an intermediate

Table 3: Classification accuracy for images with varied resolutions.

| Dataset | Image size | Trained on all resolutions | | Trained only on original resolution | | |
| --- | --- | --- | --- | --- | --- | --- |
| | | max-pooling | SAN | max-pooling | SAN | SAN-raw |
| Fashion MNIST | 14x14 | 0.8788 | **0.8810** | 0.2519 | **0.2884** | 0.2148 |
| | 22x22 | 0.8969 | **0.9064** | 0.7380 | **0.8247** | 0.4563 |
| | 28x28 | 0.9023 | **0.9111** | 0.9062 | **0.9150** | 0.8114 |
| | 42x42 | 0.9020 | **0.9033** | 0.5548 | **0.6893** | 0.3946 |
| | 56x56 | 0.8913 | **0.8966** | 0.3274 | **0.4515** | 0.3605 |
| CIFAR-10 | 16x16 | 0.5830 | **0.6167** | 0.3213 | 0.4145 | **0.4862** |
| | 26x26 | 0.6689 | **0.7037** | 0.5974 | **0.6706** | 0.4957 |
| | 32x32 | 0.6838 | **0.7292** | 0.6891 | **0.7302** | 0.4968 |
| | 48x48 | 0.6813 | **0.7080** | 0.5542 | **0.5921** | 0.4932 |
| | 64x64 | 0.6384 | **0.6413** | 0.3904 | 0.3658 | **0.4922** |

layer to apply fully connected layers. In this experiment, we compared **SAN** with **max-pooling** in classifying images of different sizes.

We used analogical architecture as in Section 4.3. Note that we were unable to use **flatten**, because the output from convolutional network had different sizes. We also would like to verify how much we gain using convolutional layers before applying SAN. Thus we additionally considered a variant, where SAN was applied directly to image data (with no convolutions), which we refer to as **SAN-raw**.

We again considered Fashion MNIST and CIFAR-10 data sets. To create examples with different sizes we used bicubic interpolation on randomly selected images[4]. We examined two cases. In the first one, the network was trained only on images with original resolution, but tested on images with different resolutions. In the second case, scaled images were used both in training and testing.

The results presented in Table 3 show that **SAN** produced more accurate results than **max-pooling** for every image resolution (compare columns 3 with 4 and 5 with 6). Observe that the results are worse when only images with $32 \times 32$ size were used in train set. It can be explained by the fact that convolutional filters were not trained to recognize relevant features from images with different scales. In this case, the difference between **SAN** and **max-pooling** are even higher.

It is evident from Table 3 that convolutional network was essential in obtaining good classification results. When no convolutions were used, the accuracy of SAN dropped significantly (last column). Another reason is that this architecture had significantly less parameters than previous convolutional network, which limited its capacity and classification performance. This also explains why SAN could not obtain as high performance as CNN network in text classification or activity prediction. More interesting thing is that, the classification accuracy in this case was similar for every image resolution, which suggests that this representation is less sensitive to the change of scale.

## 6 CONCLUSION

We proposed a general framework for processing structured data by neural network. By combining ideas from computer tomography with a natural structure of neural networks, we created set aggregation network, which learns a rich representation of complex data. It is a natural alternative for pooling operations, which squeeze a set of vectors to a single representative. Its usefulness was analyzed theoretically and justified practically in the extensive experimental study. In future, we plan to combine SAN with specialized preprocessing networks for extracting meaningful features from graph structures and text data.

---

[4]For CIFAR-10, original images of size $32 \times 32$ were scaled to $16 \times 16, 24 \times 24, 32 \times 32, 48 \times 48, 64 \times 64$. For Fashion-MNIST, images of size $28 \times 28$ were scaled to $14 \times 14, 22 \times 22, 42 \times 42, 56 \times 56$.

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

Table 4: Architecture summary for image classification, where $N$ is the size of input to the layer.

| Flatten | | | Max-pooling | | | SAN | | |
|---|---|---|---|---|---|---|---|---|
| Type | Kernel | Outputs | Type | Kernel | Outputs | Type | Kernel | Outputs |
| Conv 2d | 5x5 | 32 | Conv 2d | 5x5 | 32 | Conv 2d | 5x5 | 32 |
| Max pooling | 2x2 | | Max pooling | 2x2 | | Max pooling | 2x2 | |
| Conv 2d | 5x5 | 32 | Conv 2d | 5x5 | 32 | Conv 2d | 5x5 | 32 |
| Max pooling | 2x2 | | Max pooling | 2x2 | | Max pooling | 2x2 | |
| Conv 2d | 3x3 | 32 | Conv 2d | 3x3 | 32 | Conv 2d | 3x3 | 32 |
| Flatten | | | Max pooling | NxN | | SAN | | 128 |
| Dense | | 128 | Dense | | 128 | Dense | | 128 |
| Dropout (p=0.5) | | | Dense | | 128 | Dense | | 10 |
| Dense | | 10 | Dense | | 10 | | | |

## A  NETWORK ARCHITECTURE FOR IMAGE CLASSIFICATION

Table 4 presents the architecture used for image classification.

The CIFAR-10 dataset consists of 50000 train and 10000 test images of size 32x32 pixels, where each image belongs to one of 10 classes. The Fashion MNIST dataset contains 60000 train and 10000 test images of size 28x28 pixels from 10 classes.

We used the categorical crossentropy as the loss function and trained the models using random batches in adam optimization algorithm with the learning rate set to default $10^{-3}$. For the experiments where the network was trained only on images with original resolution and tested on images with different resolutions, we set the number of epochs to 30 with batches of 128 samples. For the experiments where scaled images were used both in training and testing, we set the number of epochs to 50 with batches of 128 samples.

## B  NETWORK ARCHITECTURE FOR TEXT CLASSIFICATION

The architecture for text classfication models consists of an trainable embedding followed by model-specific layers which project the input onto 300-dimensional space. Then a finall dense layer is applied. All models also include dropout with probability $p = 0.5$ on the penultimate layer. The architecture is summamrized in 5. We used 10000 and 5000 most frequent words for the vocabulary for the imdb and MR datasets, respectively.

The imdb dataset consists of 25000 test and 25000 train documents with the average length after tokenization of 234.76. The MR datasets consists of 10664 documents, with an average length after tokenization of 16.76. As the MR dataset does not include a pre-defined test set, we randomly select 10% of the data as the test set.

We used the categorical crossentropy as the loss function and trained the models using random batches in adam optimization algorithm. For the imdb dataset we set the number of epochs to 30, the embedding dimension to 128, the batch size to 256 and we perform a grid search on the learning rate in the set of $\{10^{-1}, 10^{-2}, 10^{-3}, 10^{-4}\}$. For the MR dataset we train for 20 epochs, with embedding dimension 32, learning rate $10^{-3}$ and we perform a grid search on the batch size in the set of $\{32, 64, 128, 256\}$. For the smaller MR dataset we also incorporate a $l2$ regularization on the weights with penalty 3.0.

Table 5: Architecture summary for text classification, where $N$ is the length of the input sequence and $h_{dim}$ is 128 for imdb and 32 for MR.

| CNN | | | SAN | | |
|---|---|---|---|---|---|
| Type | Kernel | Outputs | Type | Kernel | Outputs |
| Embedding | | $h_{dim}$ | Embedding | | $h_{dim}$ |
| Conv 1d | 3 | 100 | | | |
| Conv 1d | 4 | 100 | | | |
| Conv 1d | 5 | 100 | SAN | | 300 |
| Concatenate | | 300 | | | |
| Max pooling | N | 300 | | | |
| Dropout (p=0.5) | | | Dropout (p=0.5) | | |
| Dense | | 2 | Dense | | 2 |

