# OpenReview forum: "Deep processing of structured data"
_ICLR.cc/2019/Conference_

### Official Review · AnonReviewer3 · 2018-10-31

**Rating:** 4
**Confidence:** 3

**Review:**

The paper at hand describes an approach to enable neural networks to take arbitrary structured data (basically any data that is not easily represented as a fixed-dimensional vector) as input. The paper describes how such data can be represented as a set (e.g. a sequence is represented as a set of index + data) and then an auxiliary network called the set aggregating network (SAN)  is used to represent the data in a high dimensional latent space. In addition to the idea, the paper provides a theoretical analysis of the approach which shows that with a sufficiently high dimensional representation the network is able to learn a unique representation for each input example.

Good in this paper:
 - nicely on topic - this is truly about learning representations
 - interesting idea with some (albeit not-surprising) theoretical analysis

Not yet great in this paper:
 - the paper feels a bit premature in multiple ways, to me most importantly the experiments appear to be really rushed.
  Looking at table 1 - it is really unclear how to read this. The table is hardly explained and it would be good to actually compare the method to the state of the art. I understand that the authors care more about generality here - but it's much easier to build something generic that is very far from the state of the art than to build something that is closer to the state of the art. Also - I feel it would have been interesting to allow for a more direct comparison of the SAN results with the other methods. Similarly, in Table 2 - how far away is this from the state of the art.
- the variable size image-recognition task seems a bit artificial - I believe that scaling images to a fixed size is a reasonable idea and is well understood and also something that humans can work with. Dealing with variable size images for the purpose of not having a fixed size vector seems artificial and unnecessary - in this case here specifically the images are same size to begin with. By using SAN you loose a lot of the understanding of computer vision research of the last decade (e.g. it's clear that CNNs are a good idea - with SAN - you cannot really do that anymore) - so, I feel this experiment here doesn't add anything.

I feel these comments can probably be addressed by rethinking the experimental evaluation. At this point, I think the paper provides a nice idea with a theoretical analysis - but it doesn't provide enough experimental evidence that this works.

Minor comments:
 - p1: Analogically -> just drop the word
- p1: citation of Brin & Page -> this seems a bit out of place - yes, it is a method working on graphs, but it is not relevant in the context of the paper - to the best of my knowledge there are no neural networks in this.
- p2: where Radon - >where the Radon
- p2: "One can easily see " -> I cannot. Please ellaborate

---

### Official Review · AnonReviewer1 · 2018-11-02
**The paper "DEEP PROCESSING OF STRUCTURED DATA" is about transforming an arbitrary number of fixed-length input vectors to a fixed-length output vector for a neural network.**

**Rating:** 4
**Confidence:** 4

**Review:**

+ Results for SAN are on par with hand-crafted feature sets.
+ An easy to implement strategy to get from a set of an arbitrary number of fixed-length input vectors to one fixed-length output vector representing the set.


- The work is quite straight-forward and incremental as it basically replaces max-pooling with another function (RELU).
- Adding zero tuples (x,0) as input for the SAN is basically zero-padding. Therefore, the differences/advantages to zero padding and other pooling strategies do not become clear.
- The obvious baseline of using doc2vec for text classification is missing. This baseline would be beneficial to set the results into context.
- Reference to Schlichtkrull et al. [1] is missing who are actually doing something quite similar for graph data.


Since the work is quite straight-forward and the comparisons to related work are not sufficiently covered, I am currently leaning towards rejecting this paper.



[1] Schlichtkrull, M., Kipf, T. N., Bloem, P., van den Berg, R., Titov, I., & Welling, M. (2018, June). Modeling relational data with graph convolutional networks. In European Semantic Web Conference (pp. 593-607). Springer, Cham.

---

### Official Review · AnonReviewer2 · 2018-11-13

**Rating:** 4
**Confidence:** 3

**Review:**

Summary:

The paper argues that plain (fully connected) neural networks cannot represent structured data, e.g. sequences, graphs, etc. Specialized architectures have instead been invented for each such case, e.g. recurrent neural networks, graph networks etc. The paper then proposes to treat these structured data types as sets and propose a neural network method for converting a set into a fixed size vector, which can then be classified using plain neural networks. The method works by projecting each member of the set into M dimensions where they are passed through a ReLU and summed. The paper proves that given high enough dimensionality M, no information will be lost during this process. The paper performs experiments on graph data, text data and image data. The paper concludes "We proposed a general framework for processing structured data by neural network."

Quality:

The paper seems rushed. I think the paper has some language problems which unfortunately detract from the overall impression. "through its all one-dimensional projections". Should that be "through all its one-dimensional projections"? "Analogically". "can lead to the >lose< of meaningful information". "where (the?) Radon transform".

Clarity:

The aim and purpose of the paper is fairly clear. I think the method explanation is overly complicated. If I understand correctly, the proposed method is simply to map each set element to an M dimensional space using a linear projection followed by a ReLU and then summing the set elements. The section on how to implement it in practice further complicates the paper unnecessarily in my opinion. It's not clear why or whether the ReLU is important.

Originality:

Transforming sets into fixed size vectors is not new. See e.g. [1] and [2], which the paper does not reference or compare to.
To the papers defence I think the main idea is to map any structured data into a set, and then use a (relatively) simple method on it.

Significance:

The paper acknowledges that methods do indeed exist for the various structured data types, but claim that they are complex, and that the proposed method is a simple general alternative. As such the significance of the paper hinges on whether the proposed method is indeed simpler, and how it compares when it comes to performance. The proposed method is simple and general, but it does require that the information lost when converting the structured data to the set is encoded as features in the set elements, e.g. the sequence index is added. For images, the normalized position must be added. For graphs, the edges should be added, which interestingly is not done in the single graph experiment (I'd be curious how the edges could be added). This detracts somewhat from the claim to generality, since each structured data type must still be handled differently.

It's clear to me that the structural priors built into the specialized networks, e.g. recurrent, graph, etc. should help for these data structures. For this reason I think the proposed method will have a hard time comparing head to head. That is OK, it becomes a tradeoff between generality versus performance. Unfortunately I'm not convinced of the performance by the experiments. Specifically the proposed method is not compared head to head against the methods it proposes to replace.
 * On the graph data it is compared against various classifiers that use a fixed size, handcrafted representation. This is disingenuous in my opinion. For graph data, the natural method it proposed to replace is a graph neural network.
 * On the sequence data it is compared against a 1D convolutional neural network. The canonical sequence model is a recurrent neural network. Also for the larger IMDB dataset the performance drop against a CNN is considerable.
 * The image experiments are probably the strongest case. Here the authors make a compelling case that the SAN pooling operator is better than max-pooling. Unfortunately the authors use a dataset which have already been size normalized during pre-processing, and then un-normalize it. It'd be more convincing if the authors showed superior performance on a dataset of images of varying sizes using SAN, compared to normalizing them during pre-processing (with comparable runtime and parameter counts).

Pros:
 - Ambitious aim
 - Simple method

Cons:
 - Proposing to replace xyz methods, then not fairly comparing to them in their respective domains.
 - Not referencing relevant prior work on neural networks for sets.
 - Paper seems rushed/unfinished
 - Implicitly assumes classification/regression for entire structured data. In many cases, e.g. Named Entity Recognition, and many graph problems, the problem is to classify/regress on the individual elements, as part of the whole.

I could see the paper accepted, if sold less as the end-all solution for structured data and more as simply an improved pooling operator, with SOTA experiments to back up the claims. This would also allow the authors to use SOTA methods for pre-processing the data, e.g. RNNs, graph neural networks, etc., as they don't need to compete against every method.

(btw. instead of zero padding the batches, the authors can probably maintain a list of indices of set elements and use https://www.tensorflow.org/api_docs/python/tf/math/segment_sum)

[1] - Vinyals, Oriol, Samy Bengio, and Manjunath Kudlur. "Order matters: Sequence to sequence for sets." arXiv preprint arXiv:1511.06391 (2015).
[2] - Zaheer, Manzil, et al. "Deep sets." Advances in Neural Information Processing Systems. 2017.

---

### Meta-Review · Area_Chair1 · 2018-12-14
**Not acceptable in current form**

**Confidence:** 5
**Recommendation:** Reject

**Metareview:**

The reviewers agree this paper is not good enough for ICLR.